# Influences of the COVID-19 Pandemic on Intuitive Exercise and Physical Activity among College Students

**DOI:** 10.3390/bs12030072

**Published:** 2022-03-09

**Authors:** Alyssa L. Yon, Justine J. Reel, Lenis P. Chen-Edinboro, Melannie R. Pate, Jessica C. Reich, Linden A. Hillhouse, Rachel Kantor

**Affiliations:** College of Health and Human Services, University of North Carolina Wilmington, Wilmington, NC 28403, USA; alyyon7@gmail.com (A.L.Y.); edinborol@uncw.edu (L.P.C.-E.); patem@uncw.edu (M.R.P.); jcr2353@uncw.edu (J.C.R.); lah6974@uncw.edu (L.A.H.); rk3614@uncw.edu (R.K.)

**Keywords:** intuitive exercise, physical activity, mindfulness, COVID-19

## Abstract

The emergence of the COVID-19 pandemic has significantly impacted the health behaviors of people around the world, including their physical activity patterns. Intuitive exercise, a facet of one’s relationship with physical activity, is defined as one’s awareness, mindset (positive versus negative), and mindfulness when engaged in movement. The study’s purpose was to explore whether self-reported physical activity and psychological mindsets around exercise changed during the pandemic. College students (*n =* 216) described their relationship with exercise before and during the pandemic through anonymous completion of the Intuitive Exercise Scale (IEXS) and open-ended questions to provide in-depth contextualized responses about exercise habits. Participants reported significantly higher scores on intuitive exercise during the pandemic, such as on the Body Trust subscale (*M* = 3.43), compared to pre-pandemic levels (*M* = 3.20), *p* < 0.001. Moreover, varied themes related to physical activity were uncovered such as exercising for fun, exercise influenced by emotion, and loss of motivation to exercise. Important takeaways of the study include the diversity of responses to the pandemic (i.e., some participants reported an increase in physical activity levels and more positive exercise attitudes while others experienced the opposite), the need to promote self-care, and the need for positive coping strategies.

## 1. Introduction

The emergence and rapid spread of the coronavirus pandemic (COVID-19) and its many variants have impacted the health and well-being of people across the world. Individuals have been forced to cope with and adapt to the uncertainty that has become the “new normal”. Undoubtedly, stress resulting from COVID-19 has triggered a variety of emotional responses such as, but not limited to, fear of infection, boredom, frustration, anger, and confusion. Childcare issues, inadequate supplies, and the lack of reliable health-related information have presented additional challenges [1]. Therefore, many people reported psychological distress and problems with their mental health from the pandemic such as anxiety, stress, and depression, especially in college students [2,3,4]. Moreover, Scharmer et al. found that intolerance of uncertainty and anxiety due to COVID-19 was associated with a higher risk of mental disorders including eating disorder pathology and compulsive exercise [5].

College students are a vulnerable group during this time of adversity. Prior to the pandemic, students already faced numerous stressors such as balancing financial strain, excelling in academics, being involved in social activities, and experiencing mental health issues [6]. The current pandemic has undoubtedly changed daily routines and added additional challenges for college students’ well-being and mental health. Many students have reported their stress, anxiety, and depression levels rise due to various stressors brought on by the pandemic such as worrying about their own health, the health of loved ones, changes in living situations, sleep pattern disturbance, disruption in diet patterns, decrease in social life, and academic challenges as coursework delivery changes [7,8]. On the other hand, some researchers have theorized that as stress levels rise, physical activity tends to decrease, resulting in more sedentary behavior [9]. Likewise, people who engaged less in physical activity during COVID-19 experienced higher anxiety, depression, and stress scores compared to those performing physical activity [10].

Many recent studies have revealed an overall decrease in exercise levels as COVID-19 has restricted activity for many as they face various barriers such as the closure of gyms and recreation centers, mental health concerns, and an increase in sedentary time as more time is spent at home [11,12,13]. Conversely, findings from these studies also indicated that some individuals were shifting exercise to include more outdoor physical activity and home-based workouts. Similar to other studies on general-aged participants, college students’ physical activity and sports activity during the pandemic revealed an overall decrease in frequency, time spent exercising/duration, and intensity of physical activity [14,15,16]. The authors claimed these changes could be due to various factors such as a lack of motivation, absence of a coach, facility closures, getting bored working out at home, and lack of social interaction when exercising. It is important to note that the possibility of engaging in physical activity during the COVID-19 pandemic was influenced by each country’s restriction rules during the lockdown. A sedentary and inactive lifestyle lacking physical activity may lead to an increased risk of developing chronic diseases such as type 2 diabetes, cardiovascular disease, and cancer [17]. Furthermore, those who are consistently physically inactive have a greater risk for severe COVID-19 outcomes such as hospitalization and death [18,19].

Although much focus has been placed on the quantity of physical activity, one’s relationship to exercise has a psychological component as well. Intuitive exercise (IEX), representing one’s mindset around exercise, was developed to mirror the more familiar concept of intuitive eating (i.e., sensing one’s hunger and fullness) [20]. This concept was originally envisioned as a positive relationship with movement that could be fostered by becoming more mindful about how one’s body is feeling before, during, and after exercise. In a similar vein to intuitive eating that involves a shift in one’s psychological mindset from emotional eating to “intuitive” eating, healthy exercise would move away from emotional exercise to become more mindful and enjoyable.

Intuitive exercise involves listening to physical cues (e.g., pain, soreness, fatigue) instead of forcing oneself to follow rigid exercise routines regardless of injury or sickness [21]. One can be aware of body motions and senses while performing physical activity; feelings associated with movement can serve as cues for individuals to stop and start exercise [22]. Adopting an intuitive and positive relationship with physical activity may also address more negative exercise behaviors like over-exercise, exercise addiction, or dysfunctional exercise [23]. Mindfulness-based physical activity interventions and education programming has been shown to improve body image and reduce body dissatisfaction and desire for thinness, factors that have been shown to contribute to disordered eating [24].

For those in treatment or currently experiencing disordered eating and/or dysfunctional exercise, many are facing additional challenges brought on by the pandemic, which may lead to old unhealthy habits and heighten the recovery process. During the lockdown, those with eating disorders (ED) were more likely to have an increase in problems regulating their eating behaviors, an increase in physical activity thoughts/behaviors, more concerns about their physical appearance, and an increase in binge eating and compensatory exercise levels [25,26]. Overall, stress and anxiety from the pandemic have led to unhealthy behaviors, such as binging, compensatory exercise, body image concerns, and dietary restraint, which inhibit intuitiveness and mindfulness. Therefore, it is of great importance to analyze the effects of physical activity and intuitive exercise during this unprecedented time, associated with stress for many, to gain a greater understanding of the impact on college students’ physical activity levels and attitudes. There is some research to examine health behaviors during the pandemic; however, this research expands existing work to better understand one’s relationship and psychological mindset around exercise. Given the prevalence of COVID-19, this cross-sectional study’s purpose was to explore whether and how college students’ relationship with exercise had changed during the pandemic.

### Hypotheses

After reviewing the extant literature about its negative impacts on mental and physical behaviors, it was predicted that the pandemic would negatively impact physical activity behaviors among college students. For the relationships between intuitive exercise and attitudes and behaviors during the pandemic, it was hypothesized that higher intuitive exercise scores were associated with greater agreement with exercising more, greater agreement that one’s relationship with exercise was more positive, and less agreement that one had gained weight during the pandemic (Figure 1).

## 2. Materials and Methods

### 2.1. Sample Size Calculation

To determine the sample size necessary to reach adequate statistical power (0.80), an a priori power analysis for a paired sample *t*-test was conducted using G*Power [27]. Results of this analysis revealed that a sample size of 199 participants was sufficient to detect small effects (*d* = 0.20). Power was also considered for conducting structural equation models. Although there is disagreement amongst scholars, between 100 and 200 participants seems to be the absolute minimum for conducting such analyses [28,29,30,31,32,33].

### 2.2. Participants

A total of 216 eligible participants at a mid-sized university in the Southeastern region of the United States during the Fall 2020–Spring 2021 academic school year completed an online survey and were therefore enrolled in the study. One hundred and eighty-seven identified as female (87%), 27 identified as male (13%), and two classified their gender as other (0.93%). Age ranged from 18–73 (*M =* 21.10 years, *SD* = 4.92 years). A total of 40 first-year students (19%), 45 second-year students (21%), 58 third-year students (27%), 61 fourth-year students (28%), and 12 (6%) graduate students participated in the study.

### 2.3. Assessments

#### 2.3.1. Intuitive Exercise Scale (IEXS)

The IEXS, a 14-item instrument [23] designed to measure an individual’s relationship with exercise, was used. Characteristics such as enjoyment and diverse movement, mindfulness, and attention to one’s bodily cues were assessed (Appendix A). Participants ranked on a 5-point Likert-type scale (1 = strongly disagree to 5 = strongly agree) their perceptions of personal exercise behaviors prior to (“pre”) and during (“during”) the pandemic for items such as “I stop exercising when I am fatigued” and “I use exercise to help soothe my negative emotions”. Scores were calculated using the pre-established scoring guide to create a total score (*α*_pre_ = 0.59, *α*_during_ = 0.63) as well as scores for the subscales: Emotional Exercise, Body Trust, Exercise Rigidity, and Mindful Exercise. Emotional Exercise measures the use of physical activity to control negative emotions (*α*_pre_ = 0.90, *α*_during_ = 0.92). Body Trust (confidence) measures personal reliance on physiological cues for factors such as type, intensity, and frequency of exercise (*α*_pre_ = 0.84, *α*_during_ = 0.88). Exercise Rigidity assesses the variety of exercises in which an individual engages (*α*_pre_ = 0.89, *α*_during_ = 0.92). Mindful Exercise measures recognition of bodily cues to stop exercising (*α*_pre_ = 0.77, *α*_during_ = 0.89). After following scoring guidelines and reverse scoring requirements, higher total subscale scores indicated higher levels of intuitive exercise, which were considered positive in terms of health practices.

#### 2.3.2. Open-Ended Questions

The authors asked additional but open-ended questions concerning attitudes and behaviors towards exercise over the pandemic: “Think back to your behaviors BEFORE the Pandemic. What types of physical activity did you engage in?” “Think back to your behaviors BEFORE the Pandemic. How many days per week did you exercise?” “How have your EXERCISE patterns changed during the pandemic?” and “Anything else you wish to add regarding your food and exercise attitudes and behaviors during the pandemic?” The purpose of these questions was to allow participants to reflect and elaborate on their time spent during quarantine and determine whether and how their health behaviors had changed. A thematic analysis, a form of qualitative/descriptive analysis, was conducted to identify patterns and themes from responses to open-ended questions which allowed participants to provide more in-depth insight into their thoughts and feelings around physical activity [34]. Further, this descriptive analysis was used in exploring explanations for the survey findings [35].

#### 2.3.3. Exercise Attitudes and Behaviors during COVID-19

Participants also responded to statements created by the authors to assess exercise-related attitudes and behaviors DURING the pandemic, such as “I believe my relationship with exercise is MORE positive during the pandemic”, “I exercise MORE during the pandemic”, and “I have gained weight since the pandemic”, Participants responded to these statements using a 5-point Likert type scale (1 = strongly agree to 5 = strongly disagree). Each statement was analyzed individually. Responses were reverse coded such that higher scores indicated greater agreement with each statement.

### 2.4. Procedure

Once Institutional Review Board approval was granted for human subjects’ research to take place, electronic recruitment was conducted using gatekeepers from across the university. Participation was anonymous and the online survey, created in Qualtrics, took approximately 10 to 15 min to complete. Participation in the study was voluntary and convenience sampling was utilized. Participants who consented to participate had the option to discontinue the survey at any time. Participants received no incentives to participate.

Participants completed anonymous surveys related to their relationship with exercise (i.e., Intuitive Exercise Scale (IEXS); [23]) prior to and during the pandemic as part of a larger survey regarding changes in eating and exercise during COVID-19. This survey was available to participants from February 2021 to April 2021. Demographic questions were included regarding classroom level, ethnicity, Hispanic or Latino, gender, major, age, and credit hours taken. Scale instructions were reworded to capture pre and during pandemic health behaviors such as “For each item think back BEFORE the pandemic (pre-March 2020)” and “For each item please check the answer that best characterizes your CURRENT attitudes or behaviors during the pandemic”.

### 2.5. Quantitative Statistical Analysis

Established scoring instructions were applied to the IEXS responses [23]. All statistical analyses were conducted using R/RStudio v.1.4.1717. Each participant’s total score and subscale score results were calculated for “Pre” and “During” responses. We conducted paired samples *t*-tests to examine differences in mean IEXS scores prior to (“pre”) and during (“during”) the pandemic. Higher scores on the intuitive exercise scale represented a higher tendency toward intuitive exercise before or during COVID-19. We also utilized structural equation modeling to examine how IEXS scores, both prior to and during the pandemic, predicted attitudes related to exercise during the pandemic. As such, we conducted two separate structural equation models estimated using the *lavaan* package in RStudio v.1.4.1717. The first model used IEXS scores before the start of the pandemic to predict attitudes related to exercise during the pandemic. The second model used IEXS scores during the pandemic to predict attitudes related to exercise during the pandemic. The hypothesized structural model is described graphically in Figure 1. Circles represent latent variables whereas rectangles represent measured variables. We used maximum likelihood estimation (MI) for each of our models. No significant issues associated with multivariate normality were detected (e.g., −3 < skew < 3, −10 < kurtosis < 10). Further, there was no missing data in our sample. It is important to note that the intuitive exercise scale had poor reliability (i.e., internal consistency) for scores before (Cronbach’s alpha = 0.59) and during COVID-19 (Cronbach’s alpha = 0.63), so these results should be interpreted with caution. Due to the poor reliability of the overall intuitive exercise scale among our participants, we decided to conduct the remainder of the analyses on the subscales of the measure.

### 2.6. Qualitative Analysis

Participants’ open-ended responses were downloaded into a Microsoft Excel spreadsheet in preparation for a thematic analysis of the descriptive data. Open-ended responses to question 18 (eating), 19 (exercise), and 23 (other comments), were analyzed by five independent reviewers. Using Braun and Clarke’s Inductive Analysis Approach, the open-ended answers for the three questions were examined for patterns [36]. The analysis was completed in stages. First, each reviewer independently read through all the responses and noted any patterns. Second, major overlaps and commonalities in all the answers were discovered and coded. These codes were then made into themes and subthemes that categorized most of the data/responses. Based on the themes and subthemes established, responses were then color-coded and categorized. To establish credibility, each stage of the analysis was conducted independently. Then, the reviewers collaborated, reviewed, and discussed the results and patterns together to reach an inter-coder agreement to generate a finalized list of themes with a color key [37,38]. After coding each response by theme, counts were made for each theme. In addition, note-worthy comments and responses from participants were recognized that exemplified each theme.

## 3. Results

A total of 216 students began the survey and all (100%) completed it. Therefore, all responses were included in our data analysis. Ages ranged from 18 to 73 years (*M =* 21.10 years *SD* = 4.92 years). In addition, a female majority (86.6%) completed the study. Individuals from a wide range of majors and class levels completed the survey.

### 3.1. IEXS Analysis

Intuitive exercise was demonstrated by participants reporting greater trust in one’s body, less exercise driven by emotion, more exercise variety in types of activities chosen, and being more mindful during exercise. Intuitive exercise overall before COVID-19 (*M* = 3.30, *SD* = 0.45) was significantly lower than during COVID-19 (*M* = 3.36, *SD* = 0.47), *t*(215) = −2.24, *p* = 0.03, *d* = 0.15. Body Trust before COVID-19 (*M* = 3.20, *SD* = 0.98) was significantly lower compared to during COVID-19 (*M* = 3.43, *SD* = 0.96), *t*(251) = −3.45, *p* < 0.001, *d* = 0.23. In addition, Exercise Rigidity (exercise variety) before COVID-19 (*M* = 3.62, *SD* = 1.00) was significantly lower than during COVID-19 (*M* = 3.78, *SD* = 0.99), *t*(215) = −2.12, *p* = 0.03, *d* = 0.14. There were no differences before and during COVID-19 in Emotional Exercise or Mindful Exercise (*p*s > 0.34).

### 3.2. SEM Analysis

In addition to the IEX scale, several questions provided an opportunity to receive additional feedback on students’ exercise behaviors before and during the pandemic. After looking at the data from the additional questions created by the authors, structural equation modeling was utilized to examine how facets of intuitive exercise before and during the pandemic (as measured by the IEX) influenced attitudes about exercise and weight gain during the pandemic (as measured by the additional exercise attitudes and behavior questions). See Table 1 for detailed results.

#### 3.2.1. Intuitive Exercise before the Pandemic Predicting Exercise Attitudes and Behaviors during the Pandemic

The hypothesized model for intuitive exercise before the pandemic appeared to be a proper fit for the data. The CFI (i.e., comparative fit index, which examines the discrepancy between the data and the hypothesized model, adjusting for errors associated with sample size not addressed in the chi-squared test) was 0.97; the TLI (i.e.,Tucker-Lewis index is another model fit statistic that penalizes for model complexity (more parameters = more penalization)) was 0.96; and the RMSEA (i.e., root mean square error of approximation, is a fit index based on the non-centrality parameter) was 0.06, CI_90_[0.04, 0.07]. The Akaike information criterion (AIC) and Bayesian information criterion (BIC) were 9601.65 and 9834.55, respectively. The chi-squared model was significant, *X*^2^(101) = 167.85. However, there were no significant direct effects, meaning that intuitive exercise before the pandemic did not predict exercise attitudes and behaviors during the pandemic (*p*s > 0.15; see Figure 2). See Table 2 and Table 3 for descriptive data and correlations between variables for the first model.

#### 3.2.2. Intuitive Exercise during the Pandemic Predicting Exercise Attitudes and Behaviors during the Pandemic

The hypothesized model for intuitive exercise during the pandemic also appeared to be a good fit for the data. The CFI was 0.97; the TLI was 0.97; and the RMSEA was 0.06, CI_90_[0.04, 0.07]. The AIC and BIC were 9077.51 and 9310.41, respectively. The chi-squared model was significant, *X*^2^(101) = 168.09. See Figure 3 for a visual depiction of the results. Emotional exercise during the pandemic predicted participants’ agreement with two statements: having more emotional exercise was associated with less agreement that their relationship with exercise became more positive (β = −0.31) and that they had exercised more during the pandemic (β = −0.21). Exercise rigidity during the pandemic also predicted agreement with two statements: greater exercise rigidity (exercise variety) was associated with greater agreement that their relationship with exercise became more positive (β = 0.22) and that they had exercised more (β = 0.28) during the pandemic. Lastly, more mindful exercise predicted less agreement with the statements that participants’ relationship with exercise became more positive (β = −0.15) and that they had exercised more during the pandemic (β = −0.16). Exercise rigidity (*p* = 0.09) and mindful exercise (*p* = 0.06) were also marginally associated with participants’ perception of weight gain during the pandemic. Exercise rigidity (exercise variety) predicted less agreement with a perception of weight gain, whereas mindful exercise predicted greater agreement with a perception of weight gain. All other paths were nonsignificant (*p* > 0.25). See Table 4 and Table 5 for descriptives and correlations between variables for the second model.

### 3.3. Thematic Analysis

The descriptive findings illuminated how college students viewed their exercise patterns during the pandemic. After conducting a thematic analysis, many overlapping themes and subthemes emerged to illuminate patterns within the participants’ health-related responses to the pandemic. Themes and subthemes were established by identifying patterns of reoccurring words, phrases, and ideas at least approximately three to four times in the students’ responses. Students’ descriptions of their exercise patterns during the pandemic were classified into three major themes: increase in exercise, decrease in exercise, and no change in exercise. In addition, several subthemes emerged, which are described below. See Table 6 for overall themes and subthemes.

Increase in Exercise Theme. The first theme, engaging in more exercise, can be seen in many responses with phrases such as “started an exercise routine”, “increased time for exercise”, “more exercise from more time at home”, and “I have been more active since the pandemic has started”. Some students explained their physical activity changes as a way for them to escape the house and go outside: “I exercised more during the lock down, than now. During the lockdown going on a walk or run was the only way I was allowed out of the house, so I went on them frequently”. Another student similarly noted they “have found myself enjoying it [exercise] more as a way to get out of the house”. Some chose to enroll in marathons: “I have started training for a 1/2 marathon, so I am exercising more and more regularly”. Another student mentioned they “started a couch-5k running program”.

Variety of Exercises Subtheme. One subtheme that emerged was an increase in variety and diversity in participants’ forms of exercise. Many mentioned increased levels of home workouts as facilities were closed. A participant described the many types of workouts they now engage in: “I have incorporated HIIT workouts, weight training and cardio into my routines instead of just running as a form of exercise”. Another student stated they changed their exercise through “more variations in my routine. Before I used to run as my main form of exercise but now I do yoga, Zumba, and various ab routines from online videos”.

Exercising for Fun Subtheme. Another subtheme that appeared was people engaging in exercise as a form of fun and less for physical appearance: “I exercise more since the pandemic, but I exercise differently, I used to exercise for lacrosse but now I just exercise to stay healthy and because it makes me feel good”. Similarly, another student explained: “I started exercising for fun instead of being particular about body image”. Others had more time to discover the types of exercises they enjoy: “I have learned what exercises I like because I have had so much time to figure it out”.

More Time to Exercise Subtheme. A third subtheme discovered was people exercising simply because they had “increased time for exercise”. One common reason was due to work and classes being virtual, as can be seen in one student’s comment: “I am engaging in a lot more physical activity now. This is because I have more time with my online classes”. Another person commented, “I would say my exercise patterns have improved. Before the pandemic, I exercised a lot, but now that my classes have moved online, I find myself having more time on my hands, allowing me to lengthen the amount of time I spend exercising”. One student reported, “I had more time to exercise so I began exercising more frequently, almost every day of the week. I found increasingly difficult exercise routines that I found fun to do”. Others used the extra free time to self-reflect. One person mentioned, “I have slowly been trying to increase my exercise with extra free time and having more time to self-reflect on myself”. Students had more time to explore various types of activity: “I have started exercising slightly more and I have had time to incorporate activities like walks that I didn’t get to do before”.

Exercise Influenced by Emotions Subtheme. The last major subtheme observed was using exercise to cope with emotions. One participant revealed, “… The only times I ever exercise (now and before COVID) is when I’m stressed, depressed, angry, or frustrated. I basically only exercise to blow off steam”. Others use activity to manage their negative emotions: “I have wanted to exercise more since I cannot go out as much, and I still want to stay active and manage my negative feelings”. Others do it for the positive feelings they feel after exercising, “I exercise more than ever before as a result of the pandemic. The progress and challenge that I get from exercise has brought me a lot of positive emotions”.

Decrease in Exercise Theme. In contrast, the second major theme was people exercising less since the start of the pandemic. Many participants reported answers such as “exercising less”, “become more sedentary”, and “stopped exercising”. Many reported this decrease in physical activity due to various reasons that were then grouped into major subthemes.

Loss of Motivation Subtheme. The first major subtheme is lack/loss of motivation. The pandemic has disrupted many individuals’ exercise routines, possibly affecting motivation, as they were required to find a new way to exercise. One response read, “I don’t have motivation to exercise in my apartment because it gets too stuffy and I have no support or motivation. Going to a gym or yoga/Zumba classes used to be a big help for me but now we can’t do that”. Another student stated, “Before the pandemic, I was extremely active and in shape. At the start of the pandemic, I still continued to work out. Eventually, I lost all of my motivation to stay active”.

Fear of the Virus Subtheme. A second subtheme was the decrease in exercise due to fear of getting the virus. Many can no longer perform activities as they did pre-COVID such as going to fitness classes. One student wrote, “I exercise less because I do well with group exercises but I don’t feel comfortable exercising with others during the pandemic”; another student stated, “they vary on my mood, my fear of COVID keeps me from the gym”. Another mentioned, “…not going to the gym (because don’t want to be around people or exercise in a mask) so it’s harder to work out in a home setting and by yourself sometimes”. One described their fear of spreading the virus, “I have stopped exercising altogether- partly because I’m tired all the time partly because I don’t want to go outside and get someone sick or myself sick”.

Gym and Facility Closure Subtheme. A third subtheme observed was a decrease in exercise due to the gyms and facilities being closed. One person wrote their exercise has “… changed during the pandemic drastically, due to stay at home orders and workout facilities not being open” whereas another said, “a lot less because gyms and school campus have hour restrictions and limitations”. Others added on by indicating their complete end in physical activity, “I couldn’t go to the gym, so I just stopped working out overall”. One respondent shared, “I’ve stopped my regular workout routines at the gym (cardio and weights). I occasionally get cardio exercise, but not how I used to”.

After coding and counting the frequencies based on the major themes, 108 participants mentioned engaging in more exercise in their answers, whereas 73 participants reported engaging in less exercise. There were 36 respondents that noted their exercise levels did not change. In addition, there were 14 responses that reported a change in physical activity but did not specify an increase or decrease in activity level or intensity.

## 4. Discussion

This study explored whether and how one’s relationship with exercise changed during the COVID-19 pandemic. Overall, participants reported changes in their exercise mindset and amount since the start of the pandemic. Many college students revealed developing a more positive relationship with exercise and movement during the pandemic as represented by more intuitive exercise. This finding was somewhat unexpected given previous research indicating that stress levels result in lower rates of physical activity [9].

From these findings, it appeared that students increased their reliance on bodily cues to guide exercise behaviors (body trust). From the open-ended findings, it was discovered that extra free time in quarantine allowed many to embrace and tap into their physical body and feelings to guide how much and which types of exercise to engage in. Students support this assumption by revealing reoccurring patterns of growth in intuitiveness and insightfulness in their open responses. Many reported using this time in quarantine to discover their body’s needs to achieve a healthy, balanced lifestyle.

From the findings, it can also be speculated that students engaged in a greater variety of exercise since the pandemic started (exercise rigidity). From the descriptive findings, this idea can be supported by the open responses from the survey as many students reported having “more variation in [their] routine”, “exercise[ing] differently”, and “incorporat[ing] more forms of exercise than ever before”. Many students reported having “more time” during quarantine to explore various exercises they may enjoy and broaden their workout routines.

Engaging in exercise variety during the pandemic may be related to participants having a more positive relationship with exercise because as people engage in more types of workouts and explore creative ways to move their bodies, they are more likely to find new exercises they enjoy and therefore perform more overall physical activity. This statement is supported by research from Juvancic-Heltzel et al., who revealed that having a high variety of resistance exercise equipment compared to a low variety leads to a significant increase in enjoyment of exercise, more repetitions performed, and time spent on the exercises [39]. Therefore, people may find these novel forms of exercises to be gratifying and pleasurable instead of viewing them as routine ways to become “fit”. Likewise, a separate study on children showed that a greater variety of exercises resulted in greater exercise behaviors like repetitions, more weight, and greater liking of the exercises [40]. Exploring different ways to challenge one’s body may lead to overall improvement and strength, creating greater exercise participation. By changing up one’s workout routine, people may avoid being burned out or avoid a plateau in their physical progress as having a greater variety in exercise workouts leads to more motivation to train [41] and may overall increase physical activity levels.

During the pandemic, many college students may experience new forms of stressors impacting their emotions, mental health, and physical activity. This can be seen in existent literature and the open responses in our survey [1,7,9]. Some people may manage their emotions by exercising to control their feelings even if they do not feel like exercising, reflecting the emotional exercise subscale [23]. Lichtenstein et al. concluded that HREA (higher risk of exercise addiction) exercisers experience more psychological distress, such as depression and stress, in comparison to LREA (low risk of exercise addiction) [42]. For some, this may lead to a negative relationship with exercise (over-exercising or dysfunctional exercising) as people may overuse physical activity while dealing with mental and emotional hardships leading to potential injury. There may be a possibility of people using physical activity to manage stress levels, continually causing them to think of stress when engaging in exercise. Some may realize this relationship and choose to end the negative cycle by stopping their “stress-induced” exercise tendencies. Others may instead choose to stop exercising as a whole to cope with their emotions in alternative ways. As suggested by the Centers for Disease Control and Prevention (CDC), some may try and limit exposure to the news and social media, meditate, eat healthy meals, avoid alcohol and substance use, connect with friends, family, community, or organizations to manage stress levels [43].

In addition, students who have become more mindful about their exercise during COVID may have had the time to reflect on their overall relationship with physical activity. Some may have seen their exercise habits as negative and may have even decided to decrease their exercise levels and take a break from rigorous exercise routines. Mindful exercise may also allow people to enjoy exercising at the right levels (knowing when to stop) instead of overworking their bodies and becoming burned out.

Overall, most students showed a change in exercise habits based on a change in the environment and circumstances from the pandemic. Some adjusted to the new change and took advantage of their “new” normal by exercising more, while others did not adapt as well in terms of physical activity and exercised less. In addition, quantitative results from the additional exercise attitudes and behavior statements further revealed change in exercise levels for many participants as 103 agreed, 78 disagreed, and 35 were neutral towards the statement, “I exercise MORE during the pandemic”.

There were many themes in the student’s responses that emerged and agreed with existent literature from studies on how physical activity has decreased and sedentary behaviors increased since the pandemic [12,13,14,15,16]. Specifically, in regard to exercise levels, many participants mentioned in their responses they had decreased their physical activity due to facility closures such as the gym and group fitness classes [11,13,15]. However, other students mentioned exercising outside to escape their homes and at-home workouts [13]. There also seems to be an increase in intuitive exercise for some students as they reported increasing their variety of exercise (home workouts, Zumba, HITT, etc.), reflecting greater exercise rigidity on the intuitive exercise subscale. Students also exercised as a form of fun or a way to cope with negative emotions such as stress and anxiety, reflecting the emotional exercise subscale [23].

Findings underscored the reality that responses to the pandemic related to exercise attitude and behavior varied greatly depending on the college student. Some experienced an increase in their intuitive behaviors and began new healthy exercise habits, while others decreased their physical activity levels and experienced decreased motivation. It should be recognized that some students who were more intuitive during the pandemic seemed to engage in less physical activity and/or had a lower perception (disagreement) of a positive relationship with exercise. Therefore, this reveals a more complex relationship between intuitive exercise, physical activity levels, and relationship with exercise. Others mentioned no change in their health behaviors.

A noteworthy concern was the impact COVID-19 has had on students’ overall mental health; some students discussed a decline in their mental health as they experienced anxiety, stress, and even physical health concerns. These findings further add to prior studies on how COVID is affecting mental health such as stress, depression, and anxiety in college students [7,11], These effects can be due to pre-existing (pre-COVID) mental health concerns such as anxiety and depression faced by many college students [44] and/or various additional stressors from the pandemic such as a decrease in social life, disruption in diet patterns, changes in living conditions, economic impacts, worry about relatives being infected, and changes in daily life [7,45]. In addition, preexisting stressors experienced by college students before the pandemic, such as balancing financial strain, excelling in academics, and managing social activities, could have contributed to students’ mental health status as well [6].

### Future Perspective

There were limitations to this study. First, participants had to self-report their behaviors and may have answered in a socially desirable fashion. In addition, there may have been memory bias as participants were asked to remember before the pandemic. While this study only surveyed college students at a mid-sized university in the Southeastern region of the United States, in the future, the study could be replicated with a broader population, such as with a greater range of ages, races, and ethnicities. Having a broader sample will both allow for increased generalizability and a deeper understanding of the human response to a pandemic related to exercise. In addition, one could conduct focus groups to better understand the emerging themes from the open responses in more detail such as the impact on mental health, fear of the virus, and other themes discovered in our survey.

## 5. Conclusions

This study discovered how the global pandemic has impacted physical activity and intuitive exercise behaviors among college students. Students revealed developing a more positive mindset toward exercise during the pandemic as compared with before COVID-19. Specifically, Body Trust, Exercise Rigidity (exercise variety), and overall Intuitive exercise scores during COVID-19 were significantly higher compared to before COVID-19 scores. However, given the opportunity to elaborate on their experiences related to physical activity surrounding the pandemic, students identified varied reactions and ways that their relationship with exercise has evolved throughout the pandemic. The broad range of responses could be due to personal circumstances, such as some students being “social” exercisers; for example, the closure of gyms/facilities during the pandemic may have greatly hindered activity levels as students could no longer participate in group fitness classes. In addition, some students may have opted to limit their exposure to working out around others due to fear of contracting the virus. On the other hand, some students reported taking advantage of their extra time in the day to explore new ways to be physically active.

These findings can inform public health interventions to address adverse health behaviors, prevent potential health problems, and promote healthier and intuitive behaviors among young adults during a health crisis (i.e., global pandemic). In summary, it is important for health educators to recognize the individualized nature of a person’s health behaviors in response to the pandemic. The impact of the pandemic is complex; therefore, it is important to avoid a “one size fits all” approach. In response to these concerns, health educators may want to emphasize the importance of cultivating self-acceptance, practicing self-reflection, and developing awareness within a time of adversity.

## Figures and Tables

**Figure 1 behavsci-12-00072-f001:**
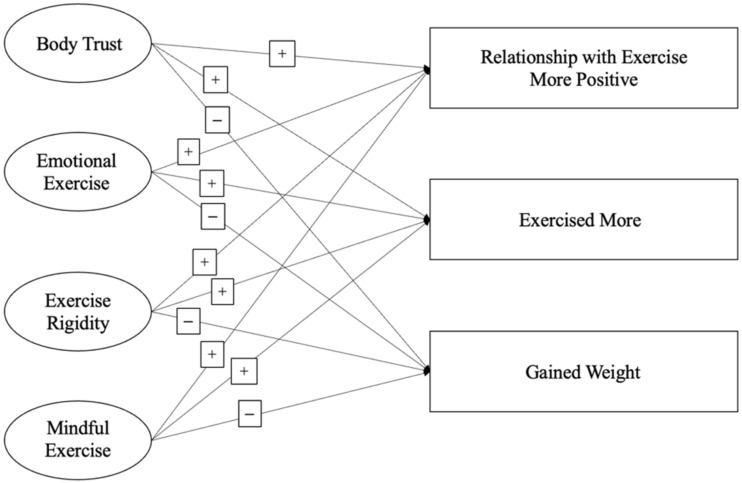
Hypothesized Structural Model for the Relationships between Intuitive Exercise and Attitudes and Behaviors During the Pandemic. Note. Emotional exercise was reverse-scored such that higher scores indicate less emotional exercise. The hypothesized structure remained the same for both intuitive exercise scores before and during the pandemic. Outcome variables always represented exercise attitudes and behaviors during the pandemic. “+” indicates a positive relationship, i.e., that higher (more positive, more intuitive) scores on each of the subscales (Body Trust, Emotional Eating, etc.) on the left are associated with higher agreement with the attitudes and behaviors on the right, whereas “−” indicates a negative relationship, i.e., that higher (more positive, more intuitive) scores on the subscales on the left are associated with lower agreement with the attitudes and behaviors on the right.

**Figure 2 behavsci-12-00072-f002:**
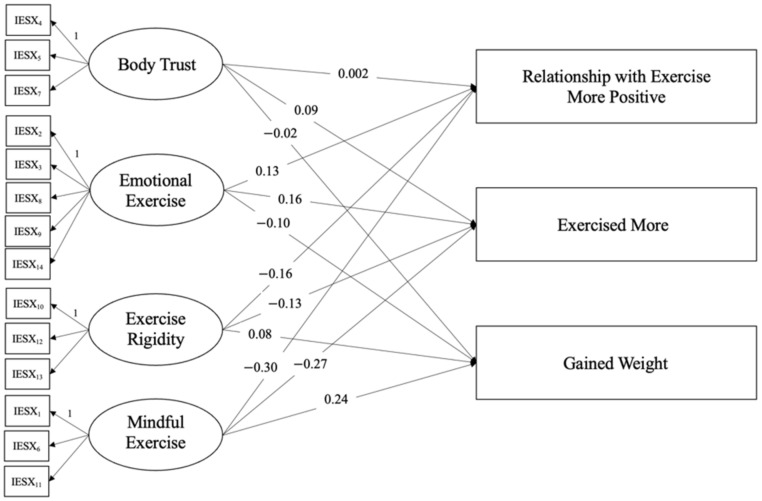
Path analysis results for facets of intuitive exercise before the pandemic predicting exercise behaviors and attitudes during the pandemic. *Note.* The first indicator for each latent variable served as the scaling indicator, represented with a “1”. No paths were significant. In this model, latent variables represented intuitive exercise before the pandemic, whereas outcomes represented exercise attitudes and behaviors during the pandemic.

**Figure 3 behavsci-12-00072-f003:**
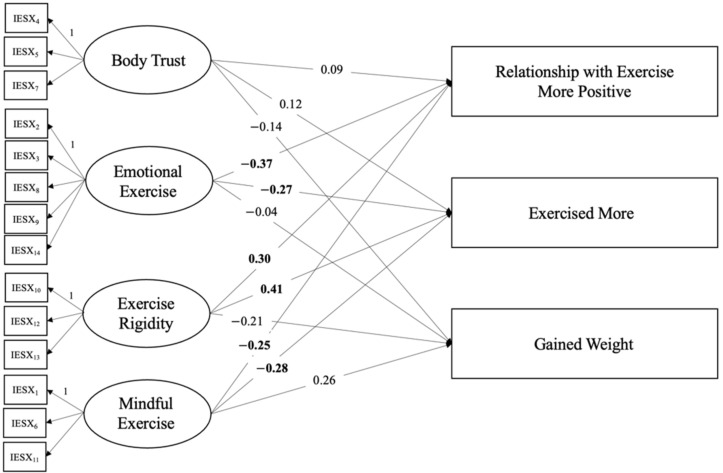
Path analysis results for facets of intuitive exercise during the pandemic and behaviors and attitudes about exercise during the pandemic. *Note.* The first indicator for each latent variable served as the scaling indicator, represented with a “1”. Coefficients in bold indicate significant paths (*p* < 0.05, two-tailed). In this model, latent variables represented intuitive exercise during the pandemic and outcome variables represented attitudes/behaviors about exercise during the pandemic.

**Table 1 behavsci-12-00072-t001:** Intuitive Exercise Before and During the Pandemic Predicting Exercise Behaviors and Attitudes during COVID-19.

Model	Before COVID-19			During COVID-19		
	β	*B*	*SE*	β	*B*	*SE*
Rel. w/Exercise More Pos ~						
BT	0.001	0.002	0.16	0.07	0.09	0.12
EE	0.10	0.13	0.14	−0.31 *	−0.37 *	0.10
ER	−0.12	−0.16	0.15	0.22 *	0.30 *	0.12
ME	−0.13	−0.30	0.21	−0.15	−0.25 *	0.13
Exercising More ~						
BT	0.05	0.09	0.17	0.08	0.12	0.12
EE	0.12	0.16	0.15	−0.21 *	−0.27 *	0.11
ER	−0.09	−0.13	0.15	0.28 *	0.41 *	0.12
ME	−0.11	−0.27	0.22	−0.16 *	−0.28 *	0.14
Gained Weight ~						
BT	−0.01	−0.02	0.15	−0.11	−0.14	0.13
EE	−0.08	−0.10	0.13	−0.04	−0.04	0.11
ER	0.06	0.08	0.14	−0.21	−0.21	0.13
ME	0.11	0.24	0.20	0.20	0.26	0.14

*Note*. Models are formatted such that outcomes are listed before the “~” and predictors are listed after. BT = Body Trust, EE = Emotional Exercise, ER = Exercise Rigidity, ME = Mindful Exercise. Rel w/ = “Relationship with”. All outcomes represented exercise attitudes and behaviors during the pandemic. * *p* < 0.05, two-tailed.

**Table 2 behavsci-12-00072-t002:** Correlations between Before COVID-19 Intuitive Exercise Latent Variables.

Variable	1		2		3		4	*M*	*SD*
	*r*	*p*	*r*	*p*	*r*	*p*	*r*		
1. BT	1	-	-	-	-	-	-	3.05	0.84
2. EE	−0.43 *	<0.001	1	-	-	-	-	2.98	1.06
3. ER	0.57 *	<0.001	−0.65 *	<0.001	1	-	-	3.47	1.00
4. ME	0.20 *	0.016	0.30 *	0.001	−0.05	0.529	1	3.68	0.56

The correlations, mean, and standard deviations are presented for the intuitive exercise latent variables before COVID-19. *Note.* BT = Body Trust, EE = Emotional Exercise, ER = Exercise Rigidity, ME = Mindful Exercise * *p* < 0.05, two-tailed.

**Table 3 behavsci-12-00072-t003:** Correlations between Observed Variables before COVID-19 Model.

Variable	1		2		3	*M*	*SD*
	*r*	*p*	*r*	*p*	*r*		
1. Rel. w/ Exercise More Positive	1	-	-	-	-	4.47	1.29
2. Exercising More	0.84 *	<0.001	1	-	-	3.87	1.37
3. Gained Weight	−0.16 *	0.022	−0.14 *	0.038	1	2.30	1.24

The correlations, means, and standard deviation values are presented for the observed variables (additional attitudes and behaviors) before COVID-19. *Note*. Rel. w/ = “Relationship with”. * *p* < 0.05, two-tailed.

**Table 4 behavsci-12-00072-t004:** Correlations between During COVID-19 Intuitive Exercise Latent Variables.

Variable	1		2		3		4	*M*	*SD*
	*r*	*p*	*r*	*p*	*r*	*p*	*r*		
1. BT	1	-	-	-	-	-	-	3.05	0.84
2. EE	−0.36 *	<0.001	1	-	-	-	-	2.98	1.06
3. ER	0.44 *	<0.001	−0.65 *	<0.001	1	-	-	3.47	1.00
4. ME	0.34 *	<0.001	0.25 *	0.002	−0.12	0.115	1	3.68	0.56

The correlations, mean, and standard deviation values are presented for the intuitive exercise latent variables during COVID-19. *Note.* BT = Body Trust, EE = Emotional Exercise, ER = Exercise Rigidity, ME = Mindful Exercise. * *p* < 0.05, two-tailed.

**Table 5 behavsci-12-00072-t005:** Correlations between Observed Variables during COVID-19 Model.

Variable	1		2		3	*M*	*SD*
	*r*	*p*	*r*	*p*	*r*		
1. Rel. w/ Exercise More Positive	1	-	-	-	-	4.47	1.29
2. Exercising More	0.79 *	<0.001	1	-	-	3.87	1.37
3. Gained Weight	−0.09	0.215	−0.06	0.414	1	2.30	1.24

The correlations, means, and standard deviation values are presented for the observed variables (additional attitudes and behaviors) during COVID-19. *Note.* Rel. w/ = “Relationship with”. * *p* < 0.05, two-tailed.

**Table 6 behavsci-12-00072-t006:** Themes and subthemes in open-ended responses.

Theme	Subtheme	Example Quotes
Increase in exercise	Variety of exercise Exercising for fun More time Emotions	“I also incorporate more forms of exercise than I have ever before”. “I started exercising for fun instead of being particular about body image”. “I have had time to incorporate activities like walks that I didn’t get to do before”. “I exercise more and use it to cope with stress”.
Decrease in exercise	Loss of motivation Fear of the virus Gym/facility closure	“Eventually, I lost all of my motivation to stay active”. “I don’t want to go outside and get someone sick or myself sick”. “Going to a gym or yoga/Zumba classes used to be a big help for me but now we can’t do that”.

## Data Availability

University of North Carolina Wilmington Randall Library has agreed to house data following acceptance of manuscript.

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
