# Peer review of "Influences of the COVID-19 Pandemic on Intuitive Exercise and Physical Activity among College Students"

_behavsci, 2022, doi:10.3390/bs12030072_

Round 1

Reviewer 1 Report

It is highly recommended to restructure the sections of the manuscript to provide a more comprehensive structure. It was hard to follow the logic of the manuscript. At the same, the authors examined a very significant topic related to evaluating the changes in physical activity during this pandemic. The use of quantitative and qualitative methods is a clear strength of the study, although its relevance could be better exploited in the study.

After the revision, it is recommended to correct the Abstract. Remove the reference from the Abstract. We believe that that are more significant results instead of the IES total score differences which is not practically significant. And in this case, it is important to note, that this significant result means a statistically significant result. Practically, this difference has not any significance (the difference between the two means is only six hundredths).

Introduction – The introduction section is a little bit too long and not focused enough. It is recommended to rewrite this section following the main purpose of the present study. This section must give an appropriate literature basis to the model that is hypothesized relationships between Intuitive Exercise (as latent variables) and Attitudes and Behaviours before and during the pandemic. (What do attitudes and behaviours relate to? this information is missing).  It is recommended to summarize the changes in physical activity and sports activity among college students during the pandemic (increased or decreased tendencies). The list of different studies, and their detailed results do not give a comprehensive picture of this field. At the end of the introduction, the authors introduce the intuitive exercise term but the significance of studying intuitive exercise during a pandemic is not highlighted enough or its presumed relationship. In Line 133 – it is recommended to give a priori hypothesis without the subscales, for example, it was hypothesized that higher intuitive exercise is associated with more exercising, a more positive relationship with exercise and not gaining weight during the pandemic. IEX abbreviation is irrelevant here because we did not get any information about the intuitive exercise scale.

Minor comments:

Line 35 – …including eating disorder pathology AND COMPULSIVE EXERCISE.

Line 37 – add some more references. There are some more relevant, up-to-date references related to changes in mental health.

Materials and Methods – The usual parts of the Materials and Methods section are quite mixed up. Please follow a logic structure – Participants, Assessments, Procedure, and Statistical Analysis with sample size considerations and the checking of the assumptions especially for SEM analysis. It is recommended to separate the quantitative and qualitative analysis sections for ease of understanding. It is recommended to move correlation tables to the Results section and to present the results of the SEM analysis in a more detailed and comprehensive way (for more details, see the comments to Results section). Please add a Statistical analysis section in detail.

Minor comments:

Sample size considerations are related to paired sample t-tests, but the authors also use SEM analysis which requires a relatively high sample size, i.e., 10 to 20 as many cases as variables.

Participants: there is a relatively high age range, or this is a mistake (18-73 yrs).

From the Materials, the presentation of the Intuitive Eating Scale is missing.

Give the meaning of the abbreviation of IEXS at the first mention.

The Cronbach alphas are very low for the total scores. But the subscales showed relatively high alphas, check the alphas for the total score because it may be miscalculated.

Remove the means and standard deviations from the paragraph, in Line 194-200, they are irrelevant in this section.

Please add the exact p-values to the correlation tables. p < 0.05 and p < 0.001 are not the standard notes for correlations (stars usually indicate the level of significance – alpha, not the p-values. But in this case, * means the correlation is significant at 0.05 level, and ** means the correlation is significant at 0.01 level).  The reported stars and p-values are correct if they are based on the exact p-values. Please check that the level of significance was not mixed up with the p-value.

Please add the meaning of the abbreviation of Rel. w/.

Results – This section contains some irrelevant paragraphs, for example, in Line 260-263, it will be more relevant to the Methods section. In Line 268-272, it is also irrelevant in the Results section. Note in the Methods section, that during the analysis the subscales were examined. It is also highly recommended to restructure this section, results related to before-during evaluation, the results of the SEM analysis, and results of the thematic analysis. It is also recommended to add subheadings to this section and separate the quantitative and qualitative analysis. It was really hard to follow the results of the present study.

Minor comments:

It is not useful to make a single-item analysis, selecting some items, especially after the subscales were examined. This paragraph did not contain any further significant information, the reported percentages are very similar; thus, reporting the most common and second common responses are irrelevant, and not informative. It is recommended to delete this paragraph and the Figures (Notes: use percentages on the figures, not frequencies data, it is confusing. At the same time, we recommended deleting these Figures).

SEM analysis – add some absolute/predictive fit indexes such as chi-squared and AIC/BIC.  What do the numbers after indexes CF1, TLI2, RMSEA3 mean? What does it mean in model 1 that “there were no significant direct effects”. The text about SEM is hard to follow, and it was difficult to understand the results of this analysis for the readers. It would be useful to put the hypothesized model here with statistical results and represent the significant paths.

Thematic analysis -it was also hard to follow the results of this thematic analysis, at the same time it is a very significant part of this manuscript. It is recommended to make a Table with the Main themes, and subthemes, and add the more common examples. In the text, summarize the characteristics and the main meaning of the themes and subthemes, it could be reported some descriptive results too.

Discussion – This section was about rather the results of the open-ended questions. It is highly recommended to restructure based on the main findings from the quantitative analysis and give further ‘evidence’ to these results from the qualitative analysis. It was the purpose of the authors as they noted in the methods. It is recommended to develop and follow a more comprehensive structure during this section, for example, based on the hypothesized model.

Minor comments

Add references to the sentence in Line 442.

Conclusions – It is recommended to move the sentence in Line 544-545 to the end of the first paragraph or delete this sentence. It sounds irrelevant after “avoid a one size fits all approach”, and it is questionable to what extent this study can be regarded as a study of mental health, or whether this can only be inferred from the results.

Reviewer 2 Report

Thanks to the authors for their submission to the Behavioral Sciences. I fully acknowledge the time and effort involved in conducting this study, the dedication to the analysis of the results, and the subsequent writing of the manuscript. However, I feel that a few points in the manuscript should be addressed by the authors.
Firstly, the introduction of the manuscript is very long and could be more concise. Also, I reckon that sentences commenting other authors’ results should be rather placed in the discussion section instead of the introduction.
Secondly, I suggest defining the type of the study.
I also recommend changing the nomenclature of the studied subgroups. I feel that terms like freshmen, sophomore, etc. are common in the US. For non-US readers, these terms may not be clear. I suggest using terms like 1st-year students, 2nd-year students, etc.
Finally, the Authors should also clarify what is the adding value of investigating such relationship in university students. The rationale behind this aim should be explored given more importance to the need of assessing this relationship from both a scientific and a practical point of view.
To sum up, the manuscript is suitable for possible publication once the suggested changes have been made.

Reviewer 3 Report

This is a paper investigating the influence of COVID-19 pandemic on intuitive exercise and physical activity in college students. The manuscript is well-written and it's of interest for the readers. However, several minor changes should be made before publishing it.

In the abstract section, I consider that the study design should be adequately described. How did the participants receive the survey or assessment measures? Was all the information complete?

The introduction section has been focused on the mental health effects of the COVID-19 pandemic and the effects on health styles. I consider very important to highlight and add two or more references about the health impact of this on medical problems. For instance, cardiovascular risk was increased? Did this increased risk impact on the occurrence of some events o diseases? Why is important to investigate the effect of physical activity? Is the relevance similar for college students compared to the other age-populations?

The possibility of doing Physical activity during the COVID-19 pandemic depends of the  restrictions of each country, almost in the lockdown. This should be stated in the introduction section.

The last paragraph of the introduction section is about aism and objectives, and about predictions. I would prefer to separate aims from Hypotheses.

The materials and methods section is very extensive. I suggest to divide it into several subsections. For instance, I would add: Participants, Sample size calculation, measures and outcomes, and statistical analyses.

The results section should be also divided into subsections according to the findings that are described. This would help the reader to better focus it.

The main conclusion is that the global COVID-19 pandemic has an impact on physical activity and exercise behaviors. However, there are many differences and reactions between participants. How are the authors explaining the varied responses and the opposite ones?

Are the authors identifying predictors of successful exercise impact or negative predictors?

A future perspectives' section is needed.
